# Investigation and Management of an Outbreak of Lead Intoxication in an Extensively Managed Beef Herd

**DOI:** 10.3390/ani13010174

**Published:** 2023-01-02

**Authors:** Meghan M. Scrivens, David Frith, Ben Wood, Brian Burren, Andrew J. Doust, Michael R. McGowan

**Affiliations:** 1School of Veterinary Science, The University of Queensland, UQ Vets Dayboro, Dayboro, QLD 4521, Australia; 2Apiam Feedlot Services, 81 Pryor St, Quirindi, NSW 2343, Australia; 3School of Veterinary Science, The University of Queensland, Gatton Campus, Lawes, QLD 4343, Australia; 4Department of Agriculture and Fisheries, Queensland Government, Coopers Plains, Brisbane, QLD 4108, Australia

**Keywords:** lead intoxication, cattle, longitudinal monitoring, management

## Abstract

**Simple Summary:**

Outbreaks of lead intoxication in grazing beef cattle are not uncommon. This report describes the outcome of the inadvertent one-week exposure of a large group of extensively managed yearling cattle to a refuse dump containing fire-damaged motor vehicle batteries. Approximately 6% of the cattle died or were euthanised between 2 to 5 weeks after exposure to the batteries. To ensure that no lead contaminated beef entered the food-chain all surviving cattle were blood sampled approximately 5 weeks after exposure. Seventy percent of cattle had no detectable lead in their blood. However, 16% of exposed cattle had blood lead concentration three times greater than the maximum normal blood lead concentration for cattle with 2% continuing to have high blood lead concentrations 1.5 years later. These cattle were euthanised and post-mortem examination revealed the presence of small pieces of lead material in their forestomach.

**Abstract:**

Fifteen hundred 12–15-month-old tropically adapted heifers inadvertently grazed a paddock which had a refuse dump in it containing burnt out vehicle batteries. The cattle grazed this paddock for approximately seven days. Subsequently these cattle were managed as two cohorts (cull and potential replacement breeding animals). Deaths commenced in the cull heifer group approximately 18 days after initial exposure to the refuse dump during relocation to a feedlot. Mortalities continued for 12 days, with other heifers showing clinical signs of marked central nervous system dysfunction requiring euthanasia. Necropsy of several clinically affected cattle plus blood sampling for lead analysis confirmed a diagnosis of lead intoxication. The crude mortality rate in the cull heifers was 6.6% (*n* = 685). Following confirmation of the diagnosis most of the potential replacement heifers (second cohort) were also relocated to the feedlot. The estimated crude mortality rate in this cohort was 5.8% (*n* = 815). All possible lead intoxication deaths occurred within 34 days of initial exposure, and apparently after day 16 at the feedlot no further heifers showed any clinical signs which could be attributed to lead intoxication. Longitudinal monitoring of blood lead concentrations was used to identify cattle suitable for slaughter. Overall, 70% of heifers initially blood sampled (*n* = 1408) had no detectable lead in their blood, however 16% had markedly elevated blood lead concentrations (> 0.7µmol/L) which persisted, and 2% had above the maximum normal threshold 1.5 years later. These latter cattle were subsequently euthanized, and necropsy revealed that visible pieces of lead were still present in the reticulum of several animals. At no time did any of these heifers with persistently high blood lead concentrations show clinical signs of lead intoxication.

## 1. Introduction

Lead intoxication is a relatively common poisoning encountered in beef cattle [1], frequently due to improper disposal of lead containing items/materials such as motor vehicle batteries and engine sump oil. However, Burren et al. [2] reported that in Australia since the introduction of a law that all new cars must use lead-free petrol, the prevalence of cases of lead intoxication in cattle has decreased. Intriguingly, when cattle are exposed to lead containing materials they seem to actively consume this material [3]. In this paper, we describe intoxication in a large herd of extensively managed beef cattle, mortalities and morbidity, and monitoring of lead levels for a period of 1.5 years following the intoxication. Published reports of lead intoxication typically involve small numbers of clinically affected animals [4,5,6] or aggregations of cases over a number of years [7]. In this case, a large number of cattle, approximately 1500, were potentially exposed and only a small proportion displayed clinical signs of lead intoxication. In the management of this case, guaranteeing food safety was the primary concern but managing the economic impact was also important.

The maximum residue limit (MRL) for lead in meat and edible offal is 0.1 mg/Kg and 0.5 mg/Kg, respectively, in accordance with the Australian New Zealand Food Standards Code and the 2015 Joint FAO/WHO Food Standards Program on Food Additives and Contaminants. For cattle, a normal blood lead concentration of less than 0.24 µmol/L (0.05 mg/L) has been adopted across Australia for investigations of suspected lead intoxication (https://www.business.qld.gov.au/_designs/content/guide-printing2?parent=78474&SQ_DESIGN_NAME=print_layout accessed on 1 November 2022). Note this is considerably lower than the reported normal blood lead concentration threshold in cattle of less than 0.96 µmol/L (0.2 mg/L) [8] and less than the 0.48 µmol/L (0.1 mg/L) threshold used in Western Canada [4]. Further, Waldner et al. [4] state that blood lead concentrations between 0.48 µmol/L and 1.7 µmol/L (0.35 mg/L) indicate some exposure to lead. Blood lead concentrations in cattle of greater than 0.5 mg/L (2.4 µmol/L) are considered conclusive evidence of lead intoxication, with cattle likely to display characteristic signs of intoxication [8]. However, a lower threshold of 1.7 µmol/L has been reported from North America [1,4]. Acute, subacute and sometimes chronic presentations of lead intoxication have been described in catttle associated with dysfunction of the central nervous and gastrointestinal systems [8]. This paper describes the investigation and management of an outbreak of lead intoxication in an extensively management beef herd in northern Australia.

## 2. Material and Methods

### 2.1. Case History

During the dry season on an extensively managed beef cattle property in northern Australia, 1500 twelve- to fifteen-month-old tropically adapted beef heifers (average weight 220 Kg) inadvertently grazed for seven days in a small paddock which had a refuse dump in it containing discarded car and truck batteries. Several years prior, a fire had burnt through this paddock melting the external casing of many of the batteries exposing the lead sheets within each. Ten days later the cattle were again walked through this paddock as they were moved to the cattle yards for drafting into two cohorts (cull and potential replacement breeding animals). One day later, 685 cull heifers (first cohort) were transported by road-train to a southern Queensland feedlot via a transit facility. No cattle were returned to the refuse dump paddock.

Within approximately 36 h of arrival at the transit facility, two heifers were found dead, and another that was recumbent was euthanized. A further heifer died during transport to the feedlot. Upon arrival at the feedlot, the heifers underwent a standard feedlot induction protocol. Within 12 days of arrival at the feedlot 45 heifers had died; of these, over half were found dead, and the remainder were euthanized as they were showing a range of clinical signs of marked central nervous system dysfunction. Affected animals displayed central nervous system signs including appearing dull to unresponsive, lateral recumbency, limb paddling and blindness, progressing to being moribund. The feedlot veterinarian was contacted, and feedlot staff were instructed to conduct necropsies on dead and euthanized heifers. Nineteen cattle were necropsied over two days; eight had plastic bags and food packaging in the rumen and in three, small pieces of dark coloured material were observed amongst the contents of the reticulum (Figure 1) and rumen (Figure 2). The latter were suspected to be pieces of lead and melted as expected when heated in the feedlot workshop; a presumptive diagnosis of lead intoxication was made. No other gross lesions were identified that could be attributed to the clinical signs observed. Testing of blood lead concentrations from two heifers that had shown neurological signs and were subsequently euthanized confirmed the diagnosis of lead intoxication (7.6 and 8.1 µmol/L). No significant hematological or biochemical changes were detected. Queensland Department of Agriculture and Fisheries (QDAF) were advised, and an animal health emergency declared with all cattle presumed to have come into contact with the refuse dump quarantined. Twenty days after arrival at the feedlot (i.e., 38 days after initial exposure to the lead contaminated refuse dump) all remaining heifers in the first cohort were bled to determine blood lead concentration. The following day all remaining heifers on the property of origin (*n* = 768; second cohort) were mustered and bled. By this time there had been 27 reported mortalities and approximately 20 heifers were missing from the second cohort of heifers.

All blood samples were collected into K_2_EDTA Vacutainer^®^ Blood Collection Tubes (Becton Dickinson Pty Ltd, Sydney, Australia). Blood lead concentrations were determined at QDAF’s Biosecurity Sciences Laboratory (Coopers Plains) according to the method described by The Australian Standard AS4090-1993X (minimum detectable concentration 0.1 µmol/L). The maximum normal blood lead threshold (MNBL) for cattle in Australia of less than 0.24 µmol/L was used as the basis for the subsequent testing and management of both cohorts of heifers. Figure 3 provides a summary of the investigation and management of this outbreak of lead exposure.

### 2.2. Animal Management and Blood Lead Concentration Monitoring

The first cohort of heifers were drafted into four management groups as follows:**Group 1:** heifers with a blood lead level below the minimum detectable concentration of 0.1 μmol/L (*n* = 479); no further testing was conducted on this group, and it was assumed this group had minimal or no contact with the lead source.**Group 2:** heifers with a blood lead level between 0.10 to 0.24 μmol/L (*n* = 42). These cattle all had detectable blood lead concentrations below or equal to the MNBL. A precautionary approach was applied, and they were retested 234 days after initial testing. At this time all had blood lead concentrations less than 0.1 µmol/L. One heifer from this group died but the death was reported to be unrelated to the outbreak of lead intoxication. The average daily weight gain of this group was 0.34 Kg/d.**Group 3:** heifers with a blood lead level > 0.24 to < 0.7 μmol/L (*n* = 32). These heifers had initial blood lead concentrations above the MNBL, but below three times the MNBL. These heifers were retested 234 days after initial testing. At this time twenty-six heifers had blood lead concentrations less than 0.1 µmol/L and six had concentrations 0.1 μmol/L to < 0.24 μmol/L. The average daily weight gain of this group was 0.33 Kg/d.**Group 4:** heifers with blood lead levels > 0.7 μmol/L (*n* = 85). These heifers all had an initial blood lead concentration ranging between 0.73 to 10.04 µmol/L, i.e., greater than three times the MNBL. There were 50 animals with blood lead concentrations of ≥1.7 µmol/L but none showed any clinical signs of illness. They were all retested 399 days after initial testing; 37 heifers had blood lead concentrations less than 0.1 µmol/L, 35 heifers had blood lead concentrations between 0.1 to < 0.24 µmol/L, 12 heifers had blood lead concentrations > 0.24 to < 0.7 μmol/L, and two heifers had blood lead concentrations > 0.7 µmol/L (maximum 0.81 μmol/L). The average daily weight gain of this group was 0.36 Kg/d. The 14 heifers with lead levels above the MNBL were tested again 94 days later; at this time all but one had blood lead concentration greater than the MNBL (range 0.23 μmol/L to 0.73 μmol/L).

The first cohort (excluding the 14 with persistent high blood lead concentration’s) were slaughtered for the Australian domestic market between 450 to 456 days after initial blood lead testing. These heifers were fed a standard feedlot ration, showed no clinical signs of lead intoxication throughout the entire feeding period, and none died before slaughter.

The second cohort of heifers (potential replacement breeding cattle) were drafted into five management groups as follows:**Group 5:** heifers with a blood lead level below the minimum detectable concentration of 0.1 μmol/L (*n* = 504), were retained on property for breeding purposes.**Group 6:** heifers with blood lead concentrations between 0.1 to <0.24 μmol/L (*n* = 67) were sent to the feedlot. Two heifers died before retesting but no cause of death was determined. The remaining heifers were retested 232 days after initial testing; 63 heifers had blood lead concentrations less than 0.1 µmol/L, and three had concentrations between 0.1 to <0.24 μmol/L. All of the Group 5 and 6 heifers were slaughtered for the Australian domestic market between 450 to 456 days after initial testing was done.**Group 7:** heifers with blood lead concentrations > 0.24 to < 0.7 μmol/L (n= 62; range 0.24 to 0.68 μmol/L), were sent to the feedlot. One heifer died with clinical signs unrelated to lead intoxication. The remaining heifers were retested 232 days after initial testing. There were 53 with blood lead levels < 0.1 μmol/L, and seven with blood lead levels between 0.1 to 0.13 μmol/L.**Group 8:** 135 heifers had blood lead concentrations > 0.7 µmol/L (range 0.71 to 9.43 μmol/L) and were relocated to a research station for long term monitoring. Further data from these heifers was not available for analysis.**Group 9**: further investigation of whether any other cattle could have been exposed to the refuse dump indicated that a group (*n* = 125) of 12- to 15-month-old tropically adapted steers also had access around the same time as the heifers. They were sent to the feedlot and underwent blood lead testing. All but two steers had blood lead concentrations less than 0.1 μmol/L. Both these steers (initial blood lead concentrations of 0.4 and 0.43 μmol/L) were retested 155 days later; the blood lead concentration of one steer had decreased to below the MNBL, however the other steer’s blood lead concentration was 0.35 μmol/L. This steer was retested a further 104 days later; at this time the blood lead concentration had increased to 0.52 μmol/L. This steer was euthanized with the 14 heifers with persistently elevated blood lead concentrations.

### 2.3. Investigation of Cattle with Persistently Elevated Blood Lead Concentrations

All cattle (14 heifers and one steer) with persistently high blood lead concentrations were euthanized 599 days after initial exposure to the fire damaged batteries. Blood samples were collected immediately for blood lead assessment, and hematology and biochemistry to determine whether there were any changes due to chronic high blood lead concentration. Testing was conducted at The University of Queensland’s Veterinary Clinical Diagnostic Pathology Laboratory, Gatton Campus. A standard field necropsy was performed on each carcass. Fresh and fixed samples of brain, heart, liver, skeletal muscle, bone marrow and unpreserved reticulum were collected. Tissue samples were frozen at −20 °C until analyzed.

All tissue testing for lead was performed at QDAF’s Biosecurity Sciences Laboratory. Tissue samples were pooled into three groups based on final blood lead concentrations: Pool one (*n* = 8), blood lead concentrations ≤ 0.3 μmol/L (0.16 to 0.3 μmol/L); Pool two (*n* = 6), blood lead concentrations > 0.3 to ≤ 0.7 μmol/L (0.36 to 0.7 μmol/L) and Pool three (*n* = 1), blood lead concentration of 14.15 μmol/L. Liver and muscle concentrations of lead were measured in mg/Kg wet weight (WW); brain samples were dried at 100 °C before being analyzed.

Twenty grams of liver and muscle were separately sub-sampled from each animal pool and homogenized. All samples were tested in triplicate with the average of the samples reported. Up to 10 brain tissue sub-samples were taken from each pool and added to an acid digestion tube to obtain a total weight of 0.1 g. An atomic absorption spectrophotometer with a graphite furnace was used for all lead analysis.

## 3. Results

### 3.1. Longitudinal Monitoring of Exposed Cattle

Assuming the four deaths that occurred during transport to the feedlot were associated with the lead intoxication event, then the overall crude mortality rate in the first cohort of heifers between departure from the property of origin (18 days after initial exposure) and slaughter (on average 491 days after initial exposure) was 6.6% (45/685). For the second cohort, if we assume that the missing heifers died of lead intoxication, then the crude mortality rate for this cohort was 5.8% (47/815). All possible lead intoxication deaths occurred within 34 days of initial exposure, and apparently after day 16 at the feedlot, no further heifers showed any clinical signs which could be attributed to lead intoxication, and on average they achieved a daily weight gain of 0.3 Kg/d.

Of the remaining first cohort blood sampled 38 days after initial exposure 75% (*n* = 640) did not have detectable lead in their blood (blood lead < 0.1 µmol/L), 7% had blood lead levels below the MNBL but when sampled 234 days later none had detectable lead in their blood; 5% of the first cohort had mildly elevated blood lead concentrations (>0.24 to < 0.7μmol/L), but 234 days later all had a blood lead concentration less than the MNBL (0.24 μmol/L). Thirteen percent of the surviving first cohort of heifers initially had significantly elevated blood lead concentrations (>3X MNBL), however 399 days later only 14 heifers (2%) had blood lead concentrations > 3X MNBL which persisted for a further 97 days in 13 animals. Overall, 82% of the surviving first cohort of heifers were very unlikely to have consumed any lead, with 18% likely to have consumed at least some lead.

Of the remaining second cohort heifers blood sampled 39 days after initial exposure 66% (*n* = 768) did not have detectable lead in their blood (blood lead < 0.1 µmol/L), 9% had blood lead levels below the MNBL, but when sampled 232 days later 94% had no detectable lead in their blood; 8% of the second cohort had mildly elevated blood lead concentrations (>0.24 to <0.7 µmol/L), but 232 days later all had a blood lead concentration less than the MNBL. However, 18% of the surviving second cohort of heifers initially had significantly elevated blood lead concentrations (>3X MNBL). Overall, 74% of the surviving second cohort of heifers were very unlikely to have consumed any lead, with 26% likely to have consumed at least some lead.

### 3.2. Necropsy of Cattle with Persistently Elevated Blood Lead Concentrations

All cattle were in very good body condition. There were no gross or histopathological abnormalities detected. However, in two heifers, small pieces of lead were detected in the contents of the reticulum.

Blood lead concentrations ranged between 0.16 μmol/L to 14.15 μmol/L. Three heifers had blood lead concentration below the MNBL. No significant hematological or biochemical changes were detected.

Average tissue lead concentrations for pools one, two, and three were: 0.11 mg/Kg WW, 0.24 mg/Kg WW and 0.13 mg/kg WW, respectively for liver, and 0.01 mg/Kg WW, 0.02 mg/Kg WW and 0.01 mg/Kg WW, respectively for muscle. The average pooled brain tissue lead concentrations were 0.7 mg/Kg, 3.1 mg/kg and 0.6 mg/Kg for pools one, two and three, respectively. Overall, the lead concentrations in pooled liver and muscle samples tested were below the Australian and New Zealand Food Standards Code thresholds, but all the pooled brain samples exceeded the threshold for edible offal. Figure 4 depicts the blood lead concentration depletion curve for the 14 first cohort heifers with persistently high blood lead concentration.

## 4. Discussion

This report describes the longitudinal outcomes and changes in blood lead concentration following the short-term exposure of a large group of grazing beef cattle (*n* = 1500) to fire-damaged disused motor vehicle batteries. Including the cattle that are likely to have died of lead intoxication, it is estimated that 27% of these cattle ingested some lead with surviving cattle posing a potential public health risk. Lead–acid batteries contain metallic lead, lead dioxide, lead sulphate and sulfuric acid. On extensively managed beef cattle properties this is the major source of lead in cases of lead intoxication. Inadvertent access to discarded sump-oil and gear-box oil from diesel vehicles is considered very low risk for lead intoxication because diesel fuels contain no lead additives [2]. Accidental exposure to lead-acid batteries is reported to be the most common cause of lead intoxication in beef cattle in North America [7] and is considered the most common cause of the problem in Australia. Cattle are susceptible to consuming lead containing material because they are innately curious animals, are attracted to salty materials, and lack oral discrimination. Rumbeiha et al. [5] reported that most incidences of lead intoxication occurred following single ingestions of large amounts of lead-containing materials. The acute lethal oral dose for lead has been reported to be 600 to 800 mg/kg BW for adult cattle [9] with often the first observed signs of an outbreak of lead intoxication being cattle found dead or recumbent at pasture. Although, for practical reasons, not all deaths observed in this outbreak could be confirmed as being due to lead intoxication, all observed and apparent mortalities occurred within 34 days of initial exposure. This is important because it is the observed occurrence of mortalities plus animal showing neurological signs that will trigger investigation of a potential lead intoxication event. Had half the animals in the exposed group not been transported to a feedlot shortly after exposure it is possible that this outbreak may have gone undetected because of the extensive nature of management of beef cattle in the rangelands of northern Australia.

The overall estimated mortality rate (6%) in this outbreak of lead intoxication is within the range reported by [4] for grazing beef cattle exposed to disused batteries in Western Canada. Although it is likely that deaths due to acute lead intoxication occurred prior to the mustering of the first cohort of heifers the timing of deaths subsequently suggest that the absorption of lead was relatively slow. The reported half-life for lead in cattle is highly variable ranging from a few days to a few thousand days which makes predicting when exposed cattle can enter the food-chain very difficult. The kinetics of lead absorption and excretion are complex and affected by factors such as diet of the animal (preweaned vs. weaned), retention of lead material in the reticulum and rumen, and the chemical form of lead ingested. Only 1–2% of ingested inorganic lead is absorbed [10], whereas lead salts, such as those found in batteries are more readily absorbed [1]. The surface area of pieces of ingested lead and adherence of ingesta material to it likely affect its rate of breakdown. Additionally, if the lead material is exposed to an acid environment (e.g in the rumen of a feedlot animal experiencing sub-clinical acidosis) some of the metallic lead is converted into a more soluble lead salt [8]. This might explain the observed increase in blood lead concentration in the feedlot steer in group 9. Rubeola et al. [5] reported a half-life as short as 48 days and as long as 2507 days for cattle on different farms in Michigan which had access to disused motor vehicle batteries. Lead is excreted through feces, urine, and milk [11], the former potentially leading to significant contamination of soil where exposed cattle are intensively managed.

Despite finding pieces of lead in the reticulum of several cattle 599 days after initial exposure to the batteries, none of these cattle showed clinical, hematological, biochemical or pathological signs of chronic lead intoxication. The longitudinal decline in blood lead concentrations indicates that breakdown and absorption of lead from the retained lead pieces is also likely to have reduced over time possible due to ingesta material attaching to the lead material.

There is some debate in the literature about the use of blood lead concentrations to identify cases of lead intoxication. Parkinson et al. [8] state that some cattle with acute lead intoxication may not have increased blood lead concentrations. Further, it is interesting to note that of the heifers (first cohort) with initial markedly elevated blood lead concentration, 59% had concentrations of ≥ 1.7 µmol/L, but none exhibited any clinical signs of lead intoxication. Additionally, although it has been reported that lead intoxication can result in anemia, none of the cattle with persistently elevated blood lead concentration in this study showed any significant hematological abnormalities.

The proactive approach adopted by property and feedlot management and attending veterinarians to manage this outbreak, in particular the serial blood sampling of potentially exposed cattle, prevented lead contaminated cattle from entering the food chain, and has provided some useful guidelines for managing future outbreaks. Retrospective analysis of the longitudinal changes in blood lead concentrations by category at time of first diagnosis demonstrated the following:A high proportion (70%) of cattle had no evidence of lead ingestion and thus could have been slaughtered for human consumption after initial blood sampling.Approximately eight months after initial testing those cattle (14%) showing some evidence of lead ingestion (>0.24 to <0.7 µmol/L) all had blood lead concentrations below the MNBL (<0.24 µmol/L) and thus could have been sent to slaughter.Of those with markedly elevated blood lead concentrations (>0.7 µmol/L) a high proportion had lead concentrations below the MNBL 13 months after initial testing and thus could have been sent to slaughter.Only a very small proportion of the cattle exposed to the disused lead batteries site had persistently elevated blood lead concentrations 13 months after initial testing. However, most of these cattle still had blood concentrations greater than MNBL 19 months after initial testing.

## 5. Conclusions

The inadvertent access of a large group of cattle to lead material found in motor vehicle batteries resulted in lead intoxication and mortality in approximately 6% of exposed cattle, and a quadrupling of the time on feed in the feedlot for exposed cattle to ensure food safety. The authors suggest the following guidelines for the management of an outbreak of lead intoxication involving disused vehicle batteries:After confirmation of an outbreak of lead intoxication all cattle that could have been exposed to the source of lead must be accurately identified and blood sampled to determine their initial blood lead concentration.Cattle with above MNBL concentration must be quarantined and blood sampled again at approximately 6 and 12 months.

## Figures and Tables

**Figure 1 animals-13-00174-f001:**
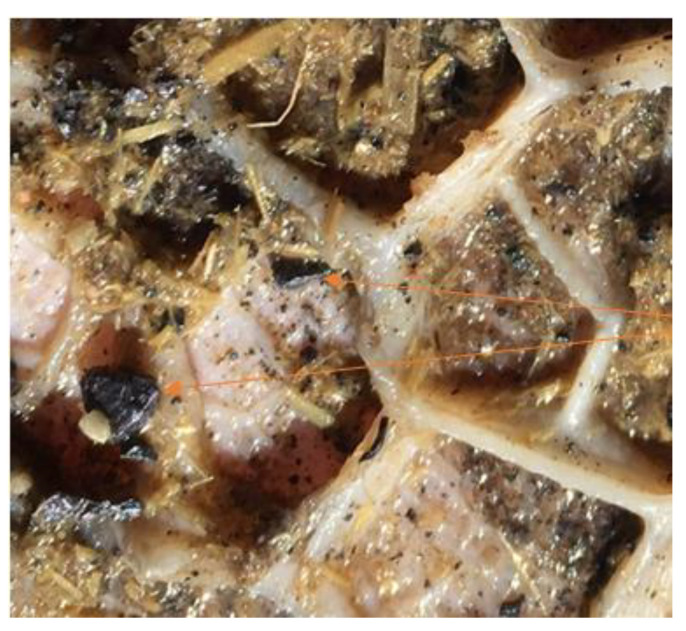
Pieces of lead (red arrows) trapped in mucosa of reticulum.

**Figure 2 animals-13-00174-f002:**
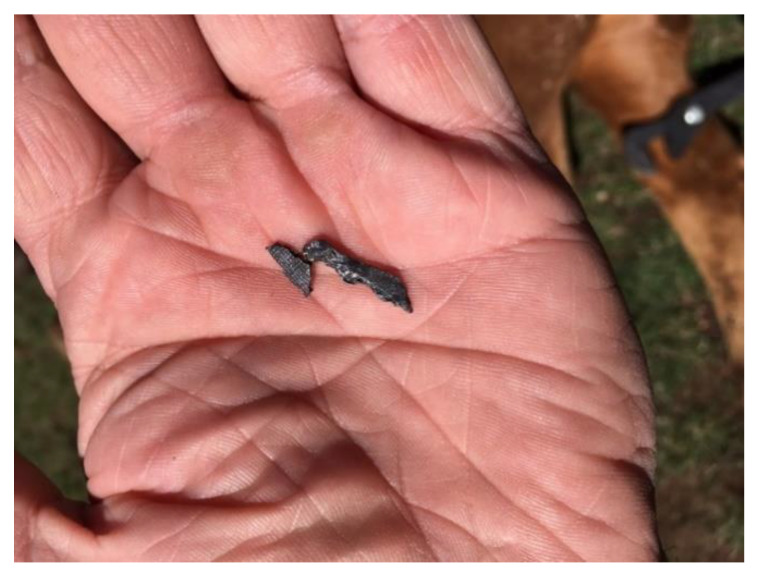
Pieces of lead removed from rumen of necropsied heifer.

**Figure 3 animals-13-00174-f003:**
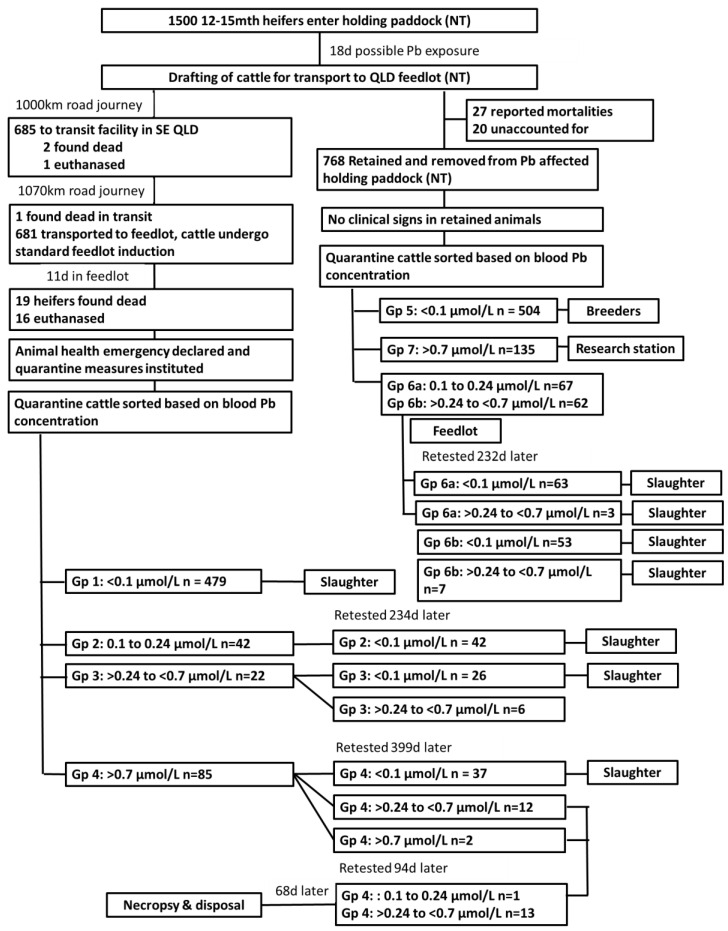
Summary of lead intoxication investigation and management.

**Figure 4 animals-13-00174-f004:**
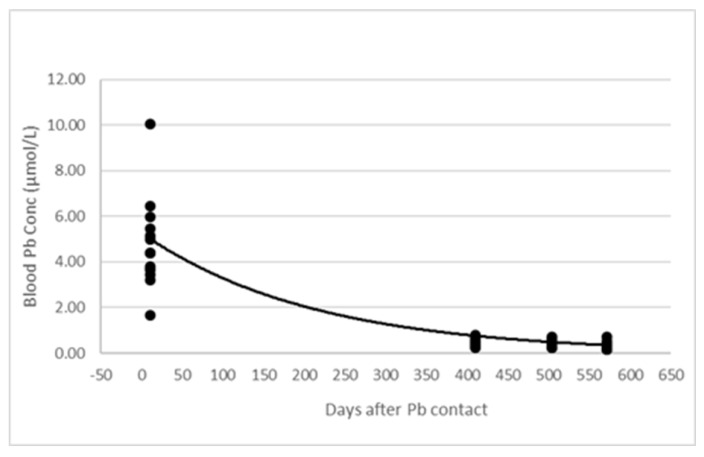
Depletion curve for blood lead concentration over 561 d for the 14 heifers.

## Data Availability

Substantial research data from this study has been presented in the paper. Access to original data is available on request.

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
