# Peer review of "Investigation and Management of an Outbreak of Lead Intoxication in an Extensively Managed Beef Herd"

_animals, 2023, doi:10.3390/ani13010174_

Round 1

Reviewer 1 Report

The paper describes and outbreak of lead poisoning with more precision that we usually are able to do. In addition to making a definitive diagnosis on the clinical and cadaverous cases, the group also dealt with cases with elevated blood lead levels and the time it took for those animals to clear the toxicosis. We don't see that very often. The report demonstrates how long it can take for levels of lead in the body to diminish to levels acceptable for food safety. What it also points out but without comment is those animals with elevated blood levels could not be sold and therefore contribute significantly to income loss of the producer.

1. Some comment on the plastic found in the GIT would round out a concern about cattle accessing a junk pile

2.  Some comment on the further performance of the heifers kept as breeders, e.g. was the conception rate within expected parameters? Would be an interesting add on.

3. Please hypothesize why in Group 9, a steer's blood lead increased from Day 155 to Day 259 post exposure.

Author Response

thank you for your feedback. Have better defined the plastic found in the rumen of necropsied cattle - added 'remnants of plastic garbage bag and food packaging'. As stated under subsection 'Group 8' regrettably the long term monitoring data for this group was not available for analysis.

With respect the observed increase in blood lead in the steer in group 9 the following has been added in the second paragraph of the Discussion:
'The surface area of pieces of ingested lead and adherence of ingesta material to it likely affect its rate of breakdown. Also if the lead material is exposed to an acid environment (e.g in the rumen of a feedlot animal experiencing sub-clinical acidosis) some of the metallic lead is converted into a more soluble lead salt. This might explain the observed increase in blood lead concentration in the feedlot steer in group 9.'  

Reviewer 2 Report

The case report "Investigation and management of an epidemic of lead poisoning in an extensively managed herd of cattle" (animals - 2080850) is well described and can be used in terms of the possibility of diagnosis and management of lead poisoning in cattle.

However, the font should be uniform throughout the manuscript (adapted to the requirements of the journal), the address for corresponding author is missing.

References cited in the text should also be corrected in accordance with the requirements of the journal.

The tests were carried out correctly, the concentration of lead in the blood of animals was determined accordingly. The division into groups was made by selecting the ranges of lead concentration in the blood of the examined cattle, justifying the division descrabded the introduction (lines 62-74). In my opinion, the authors should broaden the conclusions resulting from observations and research.

In conclusion, the manuscript may be published in the journal Animals with minor revisions.

Author Response

thankyou for your feedback. In the version of the paper sent to us for review the issue of differences in font in the text appears to have already been deal with. The citing of references has been changed in accordance with the journals requirement. With respect the conclusion the following sentence has been inserted at the beginning:

The inadvertent access of a large group of cattle to lead material found in motor vehicle batteries resulted in lead intoxication and mortality in approximately 6% of exposed cattle, and a quadrupling of the time on feed in the feedlot for exposed cattle to ensure food safety. 

Reviewer 3 Report

Let me suggest that, wanting to broaden the spectrum of fruition, one could refer to the levels allowed in other countries (e.g., the United States and Europe) and mention the main clinical symptoms that may suggest lead intoxication. Attached are the reviews in detail.

Author Response

thankyou for your feedback. The issue of differences in published minimum allowable blood lead concentration in cattle in different parts of the world is described in Ln71-76. With respect clinical signs of lead intoxication  a sentence has been added commencing ln 53 as follows:

Acute, subacute and sometimes chronic presentations of lead intoxication have been described associated with dysfunction of the central nervous and gastrointestinal systems (Parkinson et al, 2010).